# Effective Diffusion-free Score Matching for Exact Conditional Sampling

**Thomas Wedenig**[1,2]                                      **Robert Peharz**[1,2]

[1]TU Graz, Austria
[2]Graz Center for Machine Learning (GraML)

## Abstract

The success of score-based models largely stems from the idea of denoising a diffusion process given by a collection of time-indexed score fields. While diffusion-based models have achieved impressive results in sample generation, leveraging them for sound probabilistic inference—particularly for sampling from *arbitrary conditional distributions*—remains challenging. Briefly, this difficulty arises because conditioning information is only observed for clean data and not available for higher noise levels, which would be required for generating exact conditional samples. In this paper, we introduce an effective approach to *DIffusion-free SCOre matching* (DISCO), which sidesteps the need for time-dependent score fields altogether. Our method is based on a principled objective that estimates only the score of the (slightly perturbed) data distribution. In our experiments, score models learned with DISCO are competitive with state-of-the-art diffusion models in terms of sample quality. More importantly, DISCO yields a more faithful representation of the underlying data distribution and—crucially—enables sampling from arbitrary conditional distributions. This capability opens the door to sound and flexible probabilistic reasoning with score-based models.

## 1 INTRODUCTION

*Diffusion-based score models* set the current state of the art in many generative modeling tasks, producing samples of unprecedented fidelity. These models fit the score function rather than the density, waiving the need for the model's normalization constant [Hyvärinen and Dayan, 2005]. While a connection to auto-encoders leads to effective learning via *denoising score matching (DSM)* [Vincent, 2011], this objective fits the score only close to the data manifold, effectively ignoring low-density regions, leading to poor sample quality. To fix this, a key technique was the idea of generative modeling by reversing a *diffusion process* [Sohl-Dickstein et al., 2015, Song et al., 2020], which specifies a collection of time-indexed distributions. Intuitively, diffusion takes care that the model is fit on a large support, not only close to the data manifold, leading to excellent sample quality.

However, generating high-quality samples is not the only objective of probabilistic modeling. Probability theory is, at its core, a rigorous framework for reasoning under uncertainty [Jaynes, 1995, Pearl, 1988]. In particular, computing *marginals* (sum rule), which corresponds to accounting for unobserved variables, and *conditionals* (product rule), which incorporates observed evidence, are *the* fundamental operations in probabilistic reasoning [Ghahramani, 2015], and lie at the core of Bayesian methods, inverse problems and optimal decision making. Hence, the central question of this paper is: **Can score-based models serve as sound probabilistic reasoners and provide access to *exact* marginals and conditionals?** Here, we focus on drawing faithful *samples* from arbitrary marginals or conditionals, which might be used in Monte Carlo-based inference.

For marginals, one can draw samples from the joint distribution and simply discard the variables corresponding to the marginalized dimensions, yielding an exact marginal sample. However, *exact conditional sampling* is much more challenging due to the diffusion process, as conditioning the *whole* stochastic process on the available observations is intractable. Various strategies to address this problem have been proposed, which, however, are either heuristic, e.g. [Song et al., 2020, Ho et al., 2022, Kawar et al., 2022] or have only asymptotic exactness guarantees, e.g. [Wu et al., 2023]. While these methods can produce compelling results for conditional tasks such as image inpainting, they fail to produce unbiased samples from conditionals, as can be demonstrated even on simple toy problems such as in Figure 1.

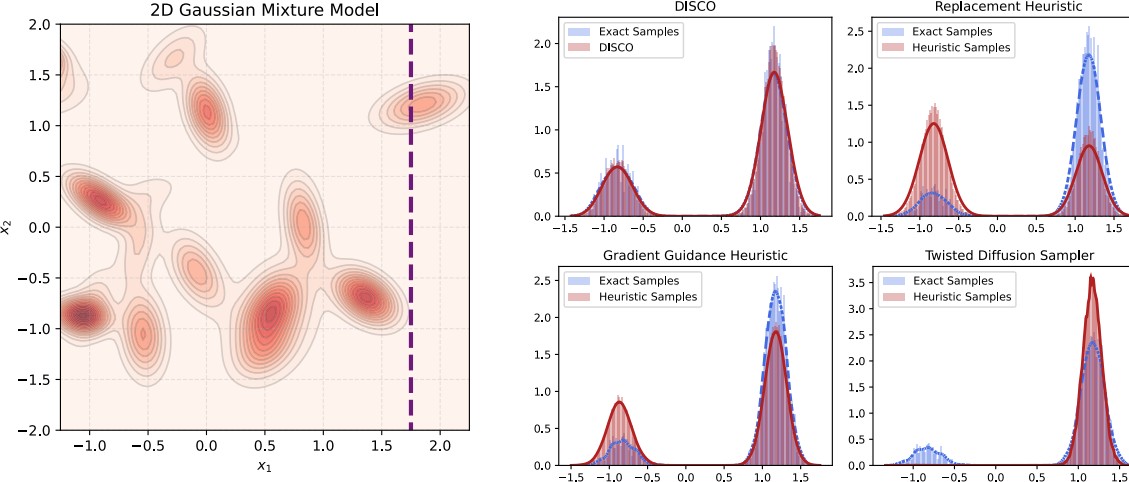

Figure 1: Conditional sampling in a low-dimensional setting. We train a score-based diffusion model and an energy-based DISCO model on samples of a 2-dimensional Gaussian mixture model ($p_d$, left). We produce conditional samples from the learned models, $x_2 \sim p_\theta(x_2 \,|\, x_1 = 1.75)$, and compare these with ground truth conditional samples derived via rejection sampling (right). For the diffusion-trained model we use *gradient guidance* [Ho et al., 2022], the *replacement heuristic* [Song et al., 2020] and *twisted SMC* [Wu et al., 2023], for which the produced samples follow a substantially different distribution than the ground truth, illustrating a clear failure case for these methods. In contrast, conditional samples from DISCO using tempered SMC follow the ground truth distribution faithfully (see supplementary for details).

In this paper, we address this problem by challenging the assumption that diffusion-based training of score-models is a pressing requirement, and propose an effective approach to *DIffusion-free SCOre matching* (DISCO). By starting from a mixture of generalized Fisher divergences, specified by an array of "noisy" proposal distributions, we arrive at a principled score matching objective. This objective, albeit reminiscent to diffusion training, only fits the (slightly perturbed) data distribution rather than a full diffusion process, while taking care that the score field is also fit outside the data manifold. With this approach, conditioning becomes simple: in the learned score, one can fix the values of observed variables and apply sampling only with respect to the unobserved variables.

In experiments, we show that DISCO produces samples of high visual quality, achieving FID scores on CIFAR-10 competitive with state-of-the-art diffusion models. More importantly, DISCO provides a more faithful representation of the underlying data distribution and enables accurate sampling from arbitrary conditional distributions, as illustrated in Figure 1. This capability opens the door to sound and flexible probabilistic reasoning with score-based models.

## 2 BACKGROUND

**Score-Based Modeling.** In generative modeling, we are given i.i.d. samples $\{\mathbf{x}^{(i)} \in \mathbb{R}^D\}_{i=1}^N$ from a data distribution $p_d(\mathbf{x})$, and aim to learn a parametric model $p_\theta$ that approximates $p_d$ well. Parameterizing a proper density $p_\theta$

introduces the challenge of normalization, i.e., ensuring that $\int_{\mathbb{R}^D} p_\theta(\mathbf{x}) \, d\mathbf{x} = 1$. Score-based modeling [Hyvärinen and Dayan, 2005] circumvents this issue by learning the *score* of the data density, defined as $\nabla_\mathbf{x} \log p_d(\mathbf{x})$, which is invariant to the normalizing constant. The idea is to use a neural network $s_\theta : \mathbb{R}^D \to \mathbb{R}^D$ to represent the model score and minimize the *Fisher divergence*:

$$\mathcal{F}(p_d \,\|\, s_\theta) := \mathbb{E}_{\mathbf{x} \sim p_d} \left[ \|\nabla_\mathbf{x} \log p_d(\mathbf{x}) - s_\theta(\mathbf{x})\|_2^2 \right] \quad (1)$$

Since the Fisher divergence involves the unknown score $\nabla_\mathbf{x} \log p_d(\mathbf{x})$, it is generally unsuitable for direct optimization. This motivates the use of alternative objectives that do not require explicit access to $\nabla_\mathbf{x} \log p_d(\mathbf{x})$. A particularly popular variant is *denoising score matching* (DSM), which approximates the score of a perturbed data distribution $p_\sigma(\tilde{\mathbf{x}}) = \int p_d(\mathbf{x}) \, q(\tilde{\mathbf{x}} \,|\, \mathbf{x}) \, d\mathbf{x}$, where $q(\tilde{\mathbf{x}} \,|\, \mathbf{x}) := \mathcal{N}(\tilde{\mathbf{x}} \,|\, \mathbf{x}, \sigma^2 I)$ is a Gaussian perturbation kernel with *fixed* noise level $\sigma$. Concretely, minimizing the objective

$$\mathcal{L}_{\text{DSM}}(\theta) := \mathbb{E}_{p_d(\mathbf{x}) q(\tilde{\mathbf{x}} \,|\, \mathbf{x})} \left[ \|\nabla_{\tilde{\mathbf{x}}} \log q(\tilde{\mathbf{x}} \,|\, \mathbf{x}) - s_\theta(\tilde{\mathbf{x}})\|_2^2 \right] \quad (2)$$

is equivalent to minimizing Fisher divergence, as $\nabla_\theta \mathcal{L}_{\text{DSM}}(\theta) = \nabla_\theta \mathcal{F}(p_\sigma \,\|\, s_\theta)$ for all $\theta$ [Vincent, 2011]. This objective and its gradients can be efficiently estimated using data samples, as it only depends on the score of the perturbation kernel, given by $\nabla_{\tilde{\mathbf{x}}} \log q(\tilde{\mathbf{x}} \,|\, \mathbf{x}) = (\mathbf{x} - \tilde{\mathbf{x}})/\sigma^2$.

In (2), one chooses the fixed noise level $\sigma$ to be small so that the perturbed distribution $p_\sigma$ closely approximates the data distribution $p_d$. However, this implies that in regions far from the data manifold, $p_\sigma$ almost never samples points,

so the learned score is essentially arbitrary there. Since sampling (e.g., via *Langevin MCMC*) typically starts far from the manifold, these inaccurate estimates lead the sampler to drift into random directions, yielding poor samples [Song and Ermon, 2019].

**Diffusion Models.** Diffusion models address the limitations of naïve DSM by learning a *multitude* of score vector fields, each corresponding to a different noise level applied to the data distribution [Sohl-Dickstein et al., 2015, Song et al., 2020]. Formally, let the clean data be denoted by $\mathbf{x}_0 \sim p_d$, and define the conditional distribution $q_t(\mathbf{x}_t \mid \mathbf{x}_0)$ via the forward diffusion process $\mathbf{x}_t = \alpha(t)\mathbf{x}_0 + \sigma(t)\boldsymbol{\varepsilon}$ where $\boldsymbol{\varepsilon} \sim \mathcal{N}(\mathbf{0}, I)$ and $t \in [0, T]$ for some $T > 0$. In this work, we focus primarily on the variance-exploding (VE) formulation [Song et al., 2020], where $\alpha(t) = 1$ and only the noise scale $\sigma(t)$ varies over time. This process defines a family of progressively noisier distributions $\{p_t(\mathbf{x}_t)\}_{t \in [0,T]}$, where $p_t(\mathbf{x}_t) = \int q_t(\mathbf{x}_t \mid \mathbf{x}_0) \, p_d(\mathbf{x}_0) \, d\mathbf{x}_0$.

A *time-dependent* score network is then trained to approximate the score function $s_{\boldsymbol{\theta}}(\mathbf{x}, t) \approx \nabla_{\mathbf{x}} \log p_t(\mathbf{x})$ for all $\mathbf{x} \in \mathbb{R}^D$ and $t \in [0, T]$, by minimizing $\mathcal{L}_{\mathrm{DM}}(\boldsymbol{\theta})$, defined as

$$\mathbb{E}_{t, \mathbf{x}_0, \mathbf{x}_t} \left[ \lambda(t) \, \| \nabla_{\mathbf{x}_t} \log p_t(\mathbf{x}_t \mid \mathbf{x}_0) - s_{\boldsymbol{\theta}}(\mathbf{x}_t, t) \|_2^2 \right] \quad (3)$$

where $t \sim p(t)$, $\mathbf{x}_0 \sim p_d(\mathbf{x}_0)$, and $\mathbf{x}_t \sim q_t(\mathbf{x}_t \mid \mathbf{x}_0)$. Here $p(t)$ is some distribution over $[0, T]$ and $\lambda(t)$ is a positive weighting function.

After training, the score network $s_{\boldsymbol{\theta}}$ is used for sample generation, aiming to approximate draws from $p_0$. Popular approaches are numerical integration of the reverse-time SDE [Song et al., 2020] and ancestral sampling [Ho et al., 2020]. A key advantage of diffusion models over standard DSM is that, due to training across multiple noise levels, the score network is also informed in low-density regions.

## 3 DIFFUSION-FREE SCORE MATCHING

While only the approximate data score at $t = 0$ is of actual interest, diffusion models introduce the overhead of an entire *family* of score functions, making conditional sampling challenging. Specifically, when splitting the data variable $\mathbf{x}$ into **u**nobserved variables $\mathbf{x}^u$ and **c**onditioned variables $\mathbf{x}^c$, the goal is to sample $\mathbf{x}^u \sim p(\mathbf{x}^u \mid \mathbf{x}^c)$. When dealing with only a *single* score field $\nabla_{\mathbf{x}} \log p(\mathbf{x})$, conditioning becomes straightforward, since the conditional score is simply the joint score with clamped $\mathbf{x}^c$:

$$\nabla_{\mathbf{x}^u} \log p(\mathbf{x}^u, \mathbf{x}^c) = \nabla_{\mathbf{x}^u} \log p(\mathbf{x}^u \mid \mathbf{x}^c) + \underbrace{\nabla_{\mathbf{x}^u} \log p(\mathbf{x}^c)}_{=0} \quad (4)$$

However, drawing conditional samples with diffusion models requires $\nabla_{\mathbf{x}_t} \log p_t(\mathbf{x}_t \mid \mathbf{x}_0^c)$ for each $t > 0$, which is intractable to compute.

In this paper, we reconsider the assumption that diffusion-based learning is strictly necessary for learning expressive score-based models. Instead, we aim to learn just a *single* score field, which allows us to sample any conditional according to (4). To this end, we start with a slight modification of the Fisher divergence:

**Definition 1.** *q-Weighted Fisher Divergence. Let $p_d$ and $q$ be probability densities over $\mathbb{R}^D$ whose supports satisfy $\mathrm{supp}(p_d) \subseteq \mathrm{supp}(q)$. We define the $q$-weighted Fisher divergence as*

$$\mathcal{F}_q(p_d \| s_{\boldsymbol{\theta}}) := \mathbb{E}_{\mathbf{x} \sim q} \left[ \| \nabla_{\mathbf{x}} \log p_d(\mathbf{x}) - s_{\boldsymbol{\theta}}(\mathbf{x}) \|_2^2 \right] \quad (5)$$

Like the Fisher divergence $\mathcal{F}$ in Equation (1), also $\mathcal{F}_q$ measures the score-mismatch between $p_d$ and the model $s_{\boldsymbol{\theta}}$, but in expectation over a *proposal distribution* $q$ rather than $p_d$. It is easy to show that $\mathcal{F}_q(p_d \| s_{\boldsymbol{\theta}}) = 0$ implies $\mathcal{F}(p_d \| s_{\boldsymbol{\theta}}) = 0$, hence $\mathcal{F}_q$ is a principled divergence.

Next, we adopt from diffusion models the idea of using a family of Gaussian perturbed distributions where $q_t(\mathbf{x}_t \mid \mathbf{x}) := \mathcal{N}(\mathbf{x}_t \mid \mathbf{x}, \sigma(t)^2 I)$ is a Gaussian perturbation kernel indexed by $t \in [0, T]$, $p_t(\mathbf{x}_t, \mathbf{x}) = q_t(\mathbf{x}_t \mid \mathbf{x}) \, p_d(\mathbf{x})$ is the joint of a data sample $\mathbf{x}$ and a perturbed version $\mathbf{x}_t$, and $p_t(\mathbf{x}_t) = \int p_t(\mathbf{x}_t, \mathbf{x}) \, d\mathbf{x}$.

Unlike as in diffusion models, we do not aim to approximate the $p_t(\mathbf{x}_t)$'s for $t > 0$, but use them merely as proposals for $\mathcal{F}_q$. We propose to minimize a *weighted mixture of $q$-weighted Fisher divergences*:

$$\mathcal{F}_{\mathrm{mix}}(p_d \| s_{\boldsymbol{\theta}}) = \mathbb{E}_{t \sim p(t)} \left[ \lambda(t) \, \mathcal{F}_{p_t}(p_d \| s_{\boldsymbol{\theta}}) \right] \quad (6)$$

$$= \mathbb{E}_{t \sim p(t)} \left[ \lambda(t) \, \mathbb{E}_{\mathbf{x}_t \sim p_t} \left[ \| \nabla_{\mathbf{x}_t} \log p_d(\mathbf{x}_t) - s_{\boldsymbol{\theta}}(\mathbf{x}_t) \|_2^2 \right] \right] \quad (7)$$

Also $\mathcal{F}_{\mathrm{mix}}$ is a principled objective, since, as $\lambda(t)$ is positive and $\mathcal{F}_{p_t}$ is non-negative, $\mathcal{F}_{\mathrm{mix}}(p_d \| s_{\boldsymbol{\theta}}) = 0$ implies that $\mathcal{F}_{p_t}(p_d \| s_{\boldsymbol{\theta}}) = 0$ for almost all $t \in [0, T]$.

$\mathcal{F}_{\mathrm{mix}}$ requires the true data score $\nabla_{\mathbf{x}} \log p_d(\mathbf{x})$ which is not available. Hence, we adopt a similar approach as in [Vincent, 2011] and replace $p_d$ with a slightly Gaussian-perturbed version $p_d'(\mathbf{x}) := p_0(\mathbf{x})$, i.e. the perturbed data distribution at the lowest noise level. Given that $\sigma(0)$ is small, fitting $p_d'$ instead of $p_d$ is a worthwhile goal. With this modification, we are able to derive the following principled objective, the *DIffusion-free SCOre matching* loss (DISCO loss):

**Theorem 1.** *Let $p_d$ be the true data distribution, $p(t)$ a distribution over $[0, T]$, and $\lambda(t)$ a positive weighting function. Further, let $q_t(\mathbf{x}_t \mid \mathbf{x})$, $p_t(\mathbf{x}_t)$ and $p_t(\mathbf{x} \mid \mathbf{x}_t)$ be defined as above. Let $p_{t'}(\mathbf{x} \mid \mathbf{x}_t) = p_{t'}(\mathbf{x}_t, \mathbf{x})/p_{t'}(\mathbf{x}_t)$ be the posterior at noise level $\sigma(t')$ and let $q(t, \mathbf{x}, \mathbf{x}_t) := p_0(\mathbf{x} \mid \mathbf{x}_t) \, p_t(\mathbf{x}_t) \, p(t)$. The DISCO loss $\mathcal{L}_{\mathrm{DISCO}}(\boldsymbol{\theta})$, defined as*

$$\mathbb{E}_{q(t, \mathbf{x}, \mathbf{x}_t)} \left[ \lambda(t) \, \| \nabla_{\mathbf{x}_t} \log q_0(\mathbf{x}_t \mid \mathbf{x}) - s_{\boldsymbol{\theta}}(\mathbf{x}_t) \|_2^2 \right] \quad (8)$$

*has the same parameter gradients as $\mathcal{F}_{\mathrm{mix}}(p_d' \| s_{\boldsymbol{\theta}})$.*

The proof can be found in the supplementary. From Theorem 1 it follows that, given that $s_{\boldsymbol{\theta}}$ has sufficient capacity, the global minimizer of $\mathcal{L}_{\text{DISCO}}$ will learn the true score of $p'_d$ and since we employ an array of noisy proposal distributions, we make sure that $s_{\boldsymbol{\theta}}$ gets informed far from the data manifold.

**DISCO Training.** Estimating $\mathcal{L}_{\text{DISCO}}$ for training is straightforward, except for one part. In order to sample from $q(t, \mathbf{x}, \mathbf{x}_t)$, we first sample $t \sim p(t)$. Subsequently, we sample $\mathbf{x}_t \sim p_t(\mathbf{x}_t)$, by first sampling some (intermediate) data sample $\mathbf{x}' \sim p_d$ and then its perturbed version $\mathbf{x}_t \sim q_t(\mathbf{x}_t \,|\, \mathbf{x}')$. The challenging part is then to sample $p_0(\mathbf{x} \,|\, \mathbf{x}_t)$.[1] However, as we usually have only finitely many training data points $\mathcal{D} = \{\mathbf{x}^{(i)}\}_{i=1}^N$, the data distribution is the empirical distribution $p_d(\mathbf{x}) = p_{\text{emp}}(\mathbf{x}) := \frac{1}{N} \sum_{i=1}^N \delta(\mathbf{x}^{(i)} - \mathbf{x})$ where $\delta(\cdot)$ denotes the Dirac-delta function. From Bayes' law, we obtain

$$p_0(\mathbf{x} \,|\, \mathbf{x}_t) = \frac{q_0(\mathbf{x}_t \,|\, \mathbf{x})\, p_{\text{emp}}(\mathbf{x})}{p_0(\mathbf{x}_t)}$$

which induces a probability mass function over $\mathcal{D}$. Thus, we compute $p_0(\mathbf{x}^{(i)} \,|\, \mathbf{x}_t) \propto q_0(\mathbf{x}_t \,|\, \mathbf{x}^{(i)})$ for each $\mathbf{x}^{(i)} \in \mathcal{D}$ and sample $\mathbf{x}$ from the normalized mass function[2].

**DISCO Samples.** If we only have access to the learned score $s_{\boldsymbol{\theta}}(\mathbf{x})$, we may use samplers like *Unadjusted Langevin Dynamics (ULA)* to draw asymptotically exact samples. However, in our low-dimensional experiments, we parameterize an energy-based DISCO model as $s_{\boldsymbol{\theta}}(\mathbf{x}) := -\nabla_{\mathbf{x}} E_{\boldsymbol{\theta}}(\mathbf{x})$, where $E_{\boldsymbol{\theta}}$ is a scalar-valued neural network. In this setting, we can employ more sophisticated MCMC sampling strategies: We use Sequential Monte Carlo (SMC) [Naesseth et al., 2019, Doucet et al., 2001, Chopin et al., 2020, Del Moral et al., 2006] with Hamiltonian Monte Carlo (HMC) steps to sample from a sequence of *tempered* distributions Neal [1996] (see supplementary for details).

## 4 EXPERIMENTS

**Low-Dimensional Experiments.** To experimentally validate DISCO in a low-dimensional setting, we train both a vanilla diffusion model and an energy-based DISCO model on a two-dimensional Gaussian mixture model (GMM). In Figure 1 we compare the quality of samples from the condi-

---

[1] Note the asymmetry in this principle, where $\mathbf{x}_t$ is generated by a perturbation at "high" noise levels, but the posterior $p_0(\mathbf{x} \,|\, \mathbf{x}_t)$ is over clean data "assuming $\mathbf{x}_t$ had been generated by $p_0$ (lowest noise level)." In particular, the intermediate sample $\mathbf{x}'$ which was used to produce $\mathbf{x}_t$ does *not* necessarily have high probability under $p_0(\mathbf{x} \,|\, \mathbf{x}_t)$, especially for large $\sigma(t)$.

[2] If $|\mathcal{D}|$ is large, we can draw an approximate posterior sample using either a mini-batch or more sophisticated techniques which are discussed in the supplementary.

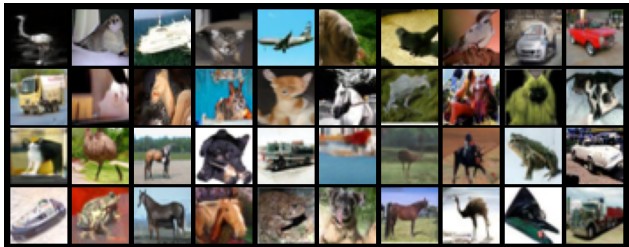

Figure 2: Unconditional samples from a score-based DISCO model trained on CIFAR-10. This model achieves an FID score of 3.80.

tional distribution $p_{\boldsymbol{\theta}}(x_2 \,|\, x_1 = 1.75)$, using popular heuristic conditional sampling techniques which are explained in the supplementary material. We find that only the DISCO model produces faithful samples, while all other methods fail to preserve the relative weights of the Gaussian components. Details and additional results for other datasets and GMMs with varying dimensionality are provided in the supplementary material.

**CIFAR-10.** To demonstrate that DISCO performs well in high-dimensional generative modeling tasks, we use the model architecture proposed in [Karras et al., 2022] and train an unconstrained score model with DISCO on the CIFAR-10 dataset [Krizhevsky et al., 2009]. Using the second-order Heun sampler Karras et al. [2022], we achieve a competitive FID score of 3.80 on unconditional CIFAR-10, where state-of-the-art with diffusion models is 1.79 [Zhang et al., 2024]. This demonstrates that directly learning a *single* data score can lead to high visual sample quality. In the supplementary, we discuss experimental details and DISCO's capability for image inpainting via conditional sampling.

## 5 CONCLUSIONS

In this paper, we challenge the prevailing belief that diffusion processes are essential for training effective score-based generative models. We introduce DISCO, a diffusion-free score matching framework that avoids time-indexed score fields in favor of learning a single, time-independent score function. Our results demonstrate that this approach is not only viable but also competitive with diffusion models in terms of visual sample quality. More importantly, DISCO provides a principled foundation for exact conditional sampling, which has remained elusive for traditional diffusion-based models. This ability opens the door to using such models as sound probabilistic reasoners, positioning DISCO as powerful tool for a wide array of tasks in probabilistic modeling, beyond mere sample generation. For example, our method might be beneficial for designing molecular structures that satisfy target binding affinities or for sampling physically plausible protein conformations conditioned on partial structural constraints.

## ACKNOWLEDGEMENTS

This project has received funding from the European Union's EIC Pathfinder Challenges 2022 programme under grant agreement No 101115317 (NEO). Views and opinions expressed are however those of the author(s) only and do not necessarily reflect those of the European Union or European Innovation Council. Neither the European Union nor the European Innovation Council can be held responsible for them.

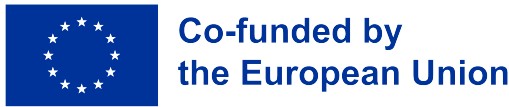

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

# Effective Diffusion-free Score Matching for
# Exact Conditional Sampling
# (Supplementary Material)

**Thomas Wedenig**[1,2]                    **Robert Peharz**[1,2]

[1]TU Graz, Austria
[2]Graz Center for Machine Learning (GraML)

## A    DISCO SAMPLING

Since it is well known that Langevin algorithms suffer from slow mixing times if the target distribution is multimodal, we employ *tempering* strategies [Neal, 1996] by considering a sequence of distributions $\{p_{\beta_i}\}_{i=0}^n$ with

$$p_{\beta_i}(\mathbf{x}) \propto p_{\boldsymbol{\theta}}(\mathbf{x})^{\beta_i} \tag{9}$$

where $0 = \beta_0 < \cdots < \beta_n = 1$ is a schedule of *inverse* temperature parameters. As $\beta \to 0$, $p_\beta$ approaches a uniform distribution, and as $\beta \to 1$, we recover the original model $p_{\boldsymbol{\theta}}$. Tempering simply scales the score, i.e., $\nabla_{\mathbf{x}} \log p_\beta(\mathbf{x}) = \beta \nabla_{\mathbf{x}} \log p_{\boldsymbol{\theta}}(\mathbf{x})$. In the same way, we can also temper any *conditional* distribution of $p_{\boldsymbol{\theta}}$ given by (4).

In our low-dimensional experiments, we use *BlackJAX* [Cabezas et al., 2024] and apply tempered sequential Monte Carlo (SMC) with an adaptive schedule for the inverse temperatures $\beta_i$.[1] For the results in Figure 1, we perform systematic resampling after a single Hamiltonian Monte Carlo (HMC) step, using 10 leapfrog integration steps. All other heuristic methods are configured to allow approximately the same number of function evaluations for a fair comparison.

## B    RELATED WORK AND CONDITIONAL SAMPLING HEURISTICS

**Time-Independence in Score-Based Models.**    Most similar in spirit to DISCO is the work by Li et al. [Li et al., 2023], who share the idea of only learning $\nabla_{\mathbf{x}} \log p_0(\mathbf{x})$ using a score-matching objective. However, they do not minimize $\mathcal{L}_{\text{DISCO}}$, but a variant which they term *multiscale denoising score matching (MDSM)*, which is $\mathcal{L}_{\text{DISCO}}$ when (incorrectly) setting $q(t, \mathbf{x}, \mathbf{x}_t) := p(t)p_d(\mathbf{x})p_t(\mathbf{x}_t \,|\, \mathbf{x})$ in (8). This objective in fact learns $s_{\boldsymbol{\theta}}^*(\mathbf{x}_t) = \mathbb{E}_{p(t\,|\,\mathbf{x}_t)}\left[\frac{\sigma(t)^2}{\sigma(0)^2}\nabla_{\mathbf{x}_t} \log p_t(\mathbf{x}_t)\right]$, i.e. a *posterior average over $p_t$ scores*, where the posterior over noise levels is reweighted. Thus, the claim of [Li et al., 2023] that $s_{\boldsymbol{\theta}}^*$ only learns the score of $p_0$ is erroneous (see Section E.2 details). Their main motivation is also not conditional sampling but on analyzing diffusion training.

A key property in DISCO is that the score network is independent of $t$, while diffusion-based models inherently rely on a notion of time. Yet, there have been attempts to minimize $\mathcal{L}_{\text{DM}}$ with neural networks where (1) time enters in a simple way, or (2) time does not enter into the network $s_{\boldsymbol{\theta}}(\mathbf{x})$ at all. [Song and Ermon, 2020] proposed to model $s_{\boldsymbol{\theta}}(\mathbf{x}, t) := \varepsilon_{\boldsymbol{\theta}}(\mathbf{x})/\sigma(t)$ where $\varepsilon_{\boldsymbol{\theta}}$ is a time-independent neural network. However, it is easy to see that the *true* scores of different noise levels are *not* just scaled versions of another, i.e., there exists no constant $c$ such that $\nabla_{\mathbf{x}} \log p_{t_1}(\mathbf{x}) = c \cdot \nabla_{\mathbf{x}} \log p_{t_2}(\mathbf{x})\ \forall \mathbf{x}, t_1 \neq t_2$, except for the trivial case where $p_0$ is Gaussian. Thus, even with infinite capacity in $\varepsilon_{\boldsymbol{\theta}}$, we cannot learn the true scores. In fact, one can interpret this parameterization as learning a single distribution whose *tempered* versions try to match the diffused distributions $p_t$. Recently, Sun et al. [2025] studied the effect of minimizing $\mathcal{L}_{\text{DM}}$ with a time-independent network $s_{\boldsymbol{\theta}}(\mathbf{x})$. Doing so results in a minimizer $s_{\boldsymbol{\theta}}^*(\mathbf{x}_t) = \mathbb{E}_{p(t\,|\,\mathbf{x}_t)}\left[\nabla_{\mathbf{x}_t} \log p_t(\mathbf{x}_t)\right]$, which learns to average the scores of $p_t$ over the posterior distribution of noise levels (see Section E.3). Sun et al. [2025] argue that in high dimensions, $p(t\,|\,\mathbf{x}_t)$ is close to $\delta(t - t_{\mathbf{x}_t})$, where $\mathbf{x}_t = \mathbf{x}_0 + t_{\mathbf{x}_t}\varepsilon, \varepsilon \sim \mathcal{N}(0, I)$, and hence, $s_{\boldsymbol{\theta}}^*(\mathbf{x}_t) \approx \nabla_{\mathbf{x}_t} \log p_{t_{\mathbf{x}_t}}(\mathbf{x}_t)$. However, this work is clearly distinct to DISCO, as we try to regress $\nabla_{\mathbf{x}} \log p_0(\mathbf{x})$ only.

---

[1]A fixed linear schedule for $\beta_i$ also performs adequately.

**Conditional Sampling in Diffusion Models.** Many approximations to the true conditional $p_0(\mathbf{x}_0^u \mid \mathbf{x}_0^c)$ have been proposed: Song et al. [2020] introduce the *replacement method*, a popular heuristic that estimates the conditional score at time $t$ as

$$\nabla_{\mathbf{x}_t^u} \log p_t(\mathbf{x}_t^u \mid \mathbf{x}_0^c) \approx \nabla_{\mathbf{x}_t^u} \log p_t(\mathbf{x}_t^u \mid \hat{\mathbf{x}}_t^c) \tag{10}$$

where $\hat{\mathbf{x}}_t^c$ is drawn from the known distribution $p_t(\mathbf{x}_t^c \mid \mathbf{x}_0^c) = \mathcal{N}(\mathbf{x}_t^c; \alpha(t)\mathbf{x}_0^c, \sigma(t)^2 I)$. This approximation enjoys no theoretical guarantees and often fails to produce samples coherent with the conditioning information [Ho et al., 2022].

*Gradient guidance* [Ho et al., 2022] relies on the fact that $\nabla_{\mathbf{x}_t} \log p_t(\mathbf{x}_t \mid \mathbf{x}_0^c) = \nabla_{\mathbf{x}_t} \log p_t(\mathbf{x}_0^c \mid \mathbf{x}_t) + \nabla_{\mathbf{x}_t} \log p_t(\mathbf{x}_t)$. While $\nabla_{\mathbf{x}_t} \log p_t(\mathbf{x}_t)$ is known via $s_{\boldsymbol{\theta}}$, the intractable quantity $p_t(\mathbf{x}_0^c \mid \mathbf{x}_t)$ is approximated, often by $\mathcal{N}(\mathbf{x}_0^c; \Omega(\hat{\mathbf{x}}_{\boldsymbol{\theta}}(\mathbf{x}_t, t)), \sigma(t)^2 I)$, where $\hat{\mathbf{x}}_{\boldsymbol{\theta}}(\mathbf{x}, t) = \mathbf{x} + \sigma(t)^2 s_{\boldsymbol{\theta}}(\mathbf{x}, t)$ is the "denoised" input, and $\Omega(\mathbf{x})$ returns only the observed coordinates in $\mathbf{x}$. At each noise level $t$, the approximation of the conditional score $\nabla_{\mathbf{x}_t} \log p_t(\mathbf{x}_t^u \mid \mathbf{x}_t^c)$ is used to perform sampling. Note that computing $\nabla_{\mathbf{x}_t} \log \mathcal{N}(\mathbf{x}_0^c; \Omega(\hat{\mathbf{x}}_{\boldsymbol{\theta}}(\mathbf{x}_t, t)), \sigma(t)^2 I)$ involves backpropagating through the neural network, making this approximation computationally expensive. Again, this heuristic provides unreliable estimates [Zhang et al., 2023] and comes with no theoretical guarantees.

[Wu et al., 2023] introduced the *twisted diffusion sampler* (TDS), which uses gradient guidance in a twisted sequential Monte Carlo (SMC) procedure as an approximation to the (unknown) optimal twisting function. Due to this, the sampler will not produce exact samples for any *finite* number of simulated particles. In contrast, DISCO guarantees asymptotically exact samples, even when simulating *a single particle*.

# C   POSTERIOR SAMPLING

When optimizing $\mathcal{L}_{\text{DISCO}}$, we need to draw samples from the $t = 0$ posterior

$$p_0(\mathbf{x} \mid \mathbf{x}_t) = \frac{q_0(\mathbf{x}_t \mid \mathbf{x}) \, p_d(\mathbf{x})}{p_0(\mathbf{x}_t)}.$$

When we set $p_d(\mathbf{x}) = p_{\text{emp}}(\mathbf{x})$, we can draw *exact* samples from $p_0(\mathbf{x} \mid \mathbf{x}_t)$: Given $\mathbf{x}_t \in \mathbb{R}^D$, we compute $q(\mathbf{x}_t \mid \mathbf{x}^{(i)})$ for each $\mathbf{x}^{(i)} \in \mathcal{D}$, and sample $\mathbf{x}$ from the normalized mass function over elements in $\mathcal{D}$. Intuitively, since the perturbation kernel $q(\mathbf{x}_t \mid \mathbf{x}^{(i)})$ is an isotropic Gaussian, it will assign more probability mass to points $\mathbf{x}^{(i)}$ that are close to $\mathbf{x}^{(i)}$. This is distinct but reminiscent of the popular (minibatch) optimal transport techniques in the flow matching literature [Tong et al., 2023].

Sampling from the posterior in this way needs $O(ND)$ operations, where $N = |\mathcal{D}|$. In our low-dimensional experiments ($N = 100,000$ and $D \in \{2, 10, 50\}$) we do not observe any significant slowdown during model training. In our high-dimensional CIFAR-10 experiments, we draw approximate posterior samples by using minibatches of size $512$.

Future work may explore utilizing techniques like *Locality Sensitive Hashing* [Gionis et al., 1999] or $k$-d Trees to efficiently get the $k$ nearest neighbors of $\mathbf{x}_t$, and then compute the mass function over just these neighbors. If $\sigma(0)$ is sufficiently small, this will be a good approximation to the true posterior mass function over all elements in $\mathcal{D}$.

# D   EXPERIMENTAL DETAILS

## D.1   LOW-DIMENSIONAL SETTING

**Moons Dataset.**   We further trained a standard diffusion model and an energy-based DISCO model on the popular *Moons* dataset (`make_moons` in `scikit-learn` [Pedregosa et al., 2011]). In Figure 3 we visualize the $L_2$ norms of learned scores (for the diffusion model the one corresponding to $t = 0$) and the score of the empirical data distribution, Gaussian smoothed with $\sigma(0) = 0.1$, which is the actual target distribution $p_d'$ for DISCO. We see that DISCO excellently fits the target score, underpinning its role as principled score matching objective. The diffusion model does not fit the data score well for areas far from the data. This is to be expected, as the diffusion formalism does not even strive to represent a single data score, but "distributes" the generative process over a hierarchy of time-dependent score-fields.

**GMM Experimental Setup.**   To quantitatively evaluate the performance of DISCO w.r.t. conditional sampling, we train several small energy-based models (parameterized using an MLP) using both $\mathcal{L}_{\text{DISCO}}$, and the regular diffusion objective $\mathcal{L}_{\text{DM}}$. For each dimension $D \in \{2, 10, 50\}$, we randomly generate parameters of a 20-component Gaussian Mixture Model (GMM) in $\mathbb{R}^D$, and use $100,000$ samples from the GMM as our training dataset.

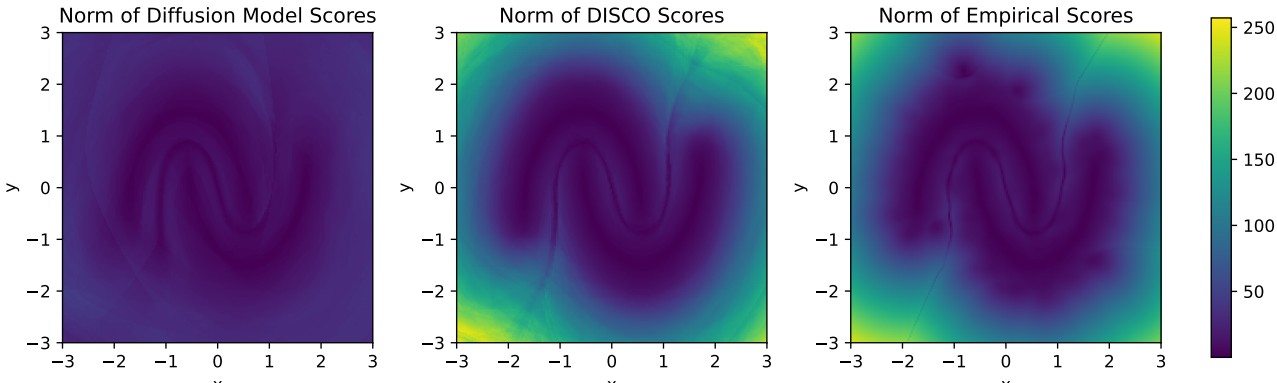

Figure 3: Comparison of the $L_2$ norms of the scores of a vanilla diffusion model at $t = 0$ (left), an energy-based DISCO model (center), and the ground truth empirical distribution of the *Moons* dataset (smoothed with $\sigma(0) = 0.1$, which is the target $p'_d$ for DISCO). Note that the diffusion model systematically underestimates the magnitudes of scores that are far from the data manifold.

We use a batch size of 1024 and the Adam optimizer [Kingma, 2014] with learning rate $10^{-4}$ and otherwise default parameters. When $D = 2$, we train the models for $50,000$ gradient steps, and when $D > 2$, we train for $100,000$ steps. We use a variance exploding formulation with $\alpha(t) = 1$ for all $t$, and use 100 exponentially spaced noise scales $\sigma(t_i)$, with $\sigma(0) = 0.1$ and $\sigma(T) = 2$. For $D = 50$, we increase $\sigma(T)$ to 5.

After training, we sample 100 test points $\mathbf{x}^{(i)}$ from the GMM, and want to draw from the conditional distribution over the last coordinate in $\mathbf{x}^{(i)}$, given the others. In the DISCO model, we use the SMC sampler described above, with systematic resampling after 2 HMC steps, which use 2 leapfrog integration steps each. To sample from the diffusion-based models, we employ several popular heuristics sampling schemes: *Twisted Diffusion Sampler (TDS)* [Wu et al., 2023], *Gradient Guidance* [Ho et al., 2022], and the *Replacement Heuristic* [Song et al., 2020]. We use all of these heuristics in conjunction with 100 steps of ancestral sampling, roughly taking the same number of function evaluations as sampling from the DISCO model[2] . For each test point, we draw 1024 (approximate) conditional samples from each model and compute the Wasserstein-1 distance ($W_1$) to 1024 true samples from the ground-truth conditional GMM. We repeat this 3 times with different random seeds and visualize the distribution of $W_1$ over all test points and random seeds in Figure 4 (for each $D \in \{2, 10, 50\}$). We find it to be beneficial for DISCO sampling to slightly lower $\sigma(0)$ after training, i.e., we sample from a slightly "cooled down" version of the learned distribution: In all DISCO experiments shown in Figure 4, we thus train with $\sigma(0) = 0.1$ and sample with $\sigma(0) = 0.07$.

As shown in Figure 4, we can easily find failure cases where methods like TDS cannot produce faithful conditional samples. In contrast, the DISCO model consistently performs well in the worst-case setting.

In the main text, we show qualitative results of a similar GMM experiment with $D = 2$, except that (1) we parameterize the diffusion model to output the score directly (instead of the energy), and (2) we show kernel-density estimates of both the approximate model conditional, and the *true model* conditional (obtained via rejection sampling). In contrast, the experiment we show here compares the approximate model conditional and the ground-truth GMM conditional.[3]

**Network Architecture.** The network architectures of the diffusion models and DISCO models are identical, except that the diffusion models receive the noise level $\sigma(t)$ as input, while the DISCO models do not. In the former case, we use a simple positional embedding for $\sigma(t)$, which we concatenate to the input. Moreover, following Tancik et al. [2020], we also use the same positional embedding for each coordinate in the input $\mathbf{x}$. The remainder of the MLP consists of 4 blocks (with residual connections), where each block contains 2 affine layers followed by leaky ReLU activations, and normalization layers at the start of each block, after the first affine layer (`InstanceNorm++` introduced in Song and Ermon [2019]). All affine layers in these blocks are maps $\mathbb{R}^K \to \mathbb{R}^K$, where we choose $K = 128$ when $D = 2$, $K = 256$ when $D = 10$, and

---

[2]Since we apply adaptive tempering when sampling from the DISCO model, the number of function evaluations per conditional sample is not static. In all experiments, the DISCO sampler needs less function evaluations than the diffusion model samplers.

[3]We make this modification because generating true samples from the model conditional via rejection sampling is intractable in high dimensions.

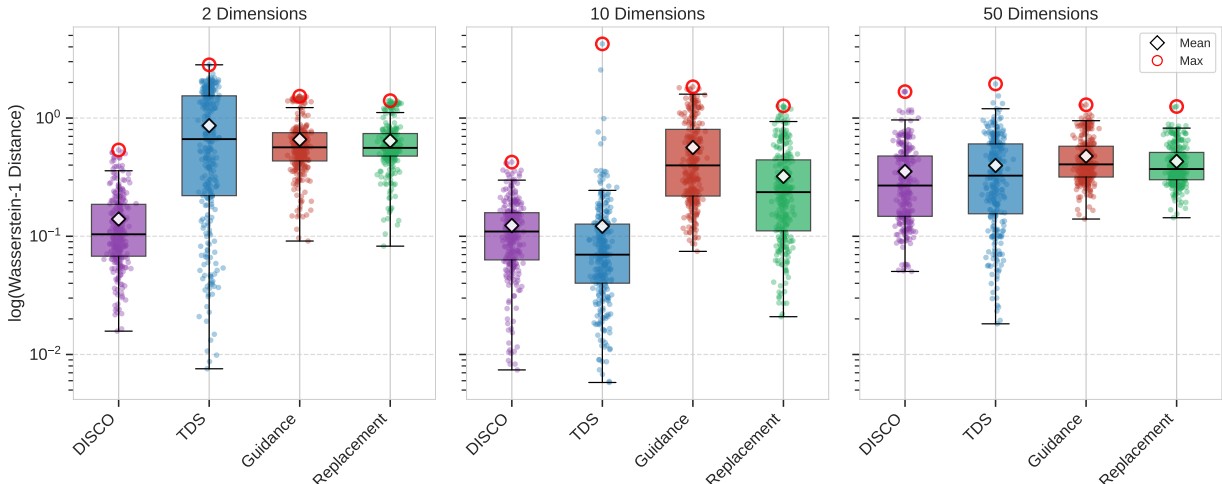

Figure 4: Wasserstein-1 distance between (approximate) conditional samples and ground-truth conditional samples for each sampling scheme and dimension $D \in \{2, 10, 50\}$. When $D = 2$ (**left**), DISCO substantially outperforms all other approaches. When $D = 10$ (**middle**), TDS performs well *on average*, but fails on certain conditionals: When using TDS, the maximum $W_1$ observed was 4.29, while the maximum $W_1$ was 0.42 with DISCO. We emphasize that such failure cases are detrimental for sound and consistent reasoning. When $D = 50$ (**right**), DISCO again outperforms all other methods in terms of average $W_1$. Note that since we compare against the ground-truth GMM conditional distribution, the errors shown here are due to sampling inaccuracy *and* model mismatch, where the latter is typically larger in higher dimensions.

$K = 512$ when $D = 50$. The final block is followed by the same normalization and activation layers, and a final affine layer mapping from $\mathbb{R}^K$ to $\mathbb{R}^D$. When parameterizing the score directly, we use the output of the final hidden layer $\mathbf{z}$ as our score approximation. When building an energy-based model, we follow Du et al. [2023] and compute the energy $E_{\boldsymbol{\theta}}$ as $-\|\mathbf{z}\|_2^2$.

## D.2 CIFAR-10

We use the popular *EDM* implementation[4] [Karras et al., 2022] which defines a *denoising network* $D_{\boldsymbol{\theta}}(\mathbf{x})$, where the score network is then given as

$$s_{\boldsymbol{\theta}}(\mathbf{x}) := \frac{D_{\boldsymbol{\theta}}(\mathbf{x}) - \mathbf{x}}{\sigma(0)^2} \tag{11}$$

Since $\nabla_{\mathbf{x}_t} \log q_0(\mathbf{x}_t \mid \mathbf{x}_0) = (\mathbf{x}_0 - \mathbf{x}_t)/\sigma(0)^2$, it follows that $\mathcal{L}_{\text{DISCO}}$ then simplifies to

$$\mathcal{L}_{\text{DISCO}}(\boldsymbol{\theta}) = \sigma(0)^{-4} \, \mathbb{E}_{q(t,\mathbf{x},\mathbf{x}_t)} \left[ \lambda(t) \|\mathbf{x} - D_{\boldsymbol{\theta}}(\mathbf{x}_t)\|_2^2 \right] \tag{12}$$

where we simply drop $\sigma(0)^{-4}$ because it is a constant factor w.r.t. $\boldsymbol{\theta}$. Karras et al. [2022] model their time-dependent denoiser as

$$D_{\boldsymbol{\theta}}(\mathbf{x}, t) := c_{\text{skip}}(t)\mathbf{x} + c_{\text{out}}(t)F_{\boldsymbol{\theta}}(c_{\text{in}}(t)\mathbf{x}, c_{\text{noise}}(t)) \tag{13}$$

where $F_{\boldsymbol{\theta}}(\cdot, \cdot)$ is the direct output of the neural network, and $c_{\text{skip}}, c_{\text{out}}, c_{\text{in}}, c_{\text{noise}}$ are scalar-valued functions. Inspired by [Sun et al., 2025], we choose time-independent constants $c_{\text{skip}} = 0.5$, $c_{\text{out}} = c_{\text{in}} = 1$, and do not use $c_{\text{noise}}$ because we model a time-independent network. Finally, we train $D_{\boldsymbol{\theta}}(\mathbf{x}) = 0.5\mathbf{x} + F_{\boldsymbol{\theta}}(\mathbf{x})$ with the same hyperparameter configuration as Karras et al. [2022][5] on 8 Quadro RTX 8000 GPUs, which took roughly 2 days. To minimize $\mathcal{L}_{\text{DISCO}}$, we approximately sample from the posterior $p_0(\mathbf{x} \mid \mathbf{x}_t)$ using a mini-batch of data.

We use the second-order Heun sampler with 18 steps (i.e., NFE = 35) [Karras et al., 2022] to produce the samples shown in Figure 2, achieving an FID score of 3.80. We also experiment with energy-based DISCO models, but observe worse visual fidelity in the generated samples, which is consistent with findings in [Salimans and Ho, 2021].

---

[4] https://github.com/NVlabs/edm

[5] To be exact, except for the discussed changes, we use the configuration of their `cifar10-32x32-uncond-vp` model.

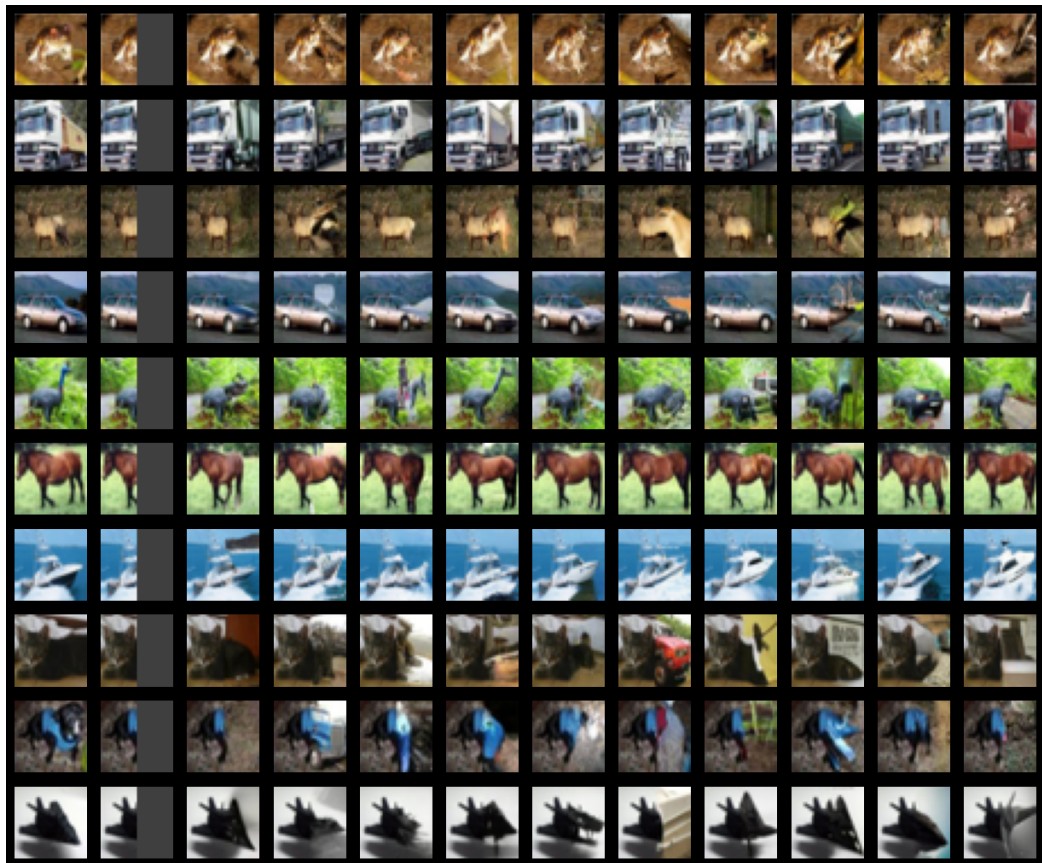

Figure 5: CIFAR-10 Inpainting with the DISCO model. **First Column**: CIFAR-10 test image from each class. **Second Column**: Pixels from the test images we condition on. **Other Columns**: Inpaintings with different random seeds using the DISCO model in conjunction with the replacement heuristic. Best viewed zoomed in.

**CIFAR-10 Inpainting.**  We use the DISCO model described in the main paper in an inpainting experiment. The results are shown in Figure 5: In this experiment, we use the popular replacement heuristic to inpaint images. We find that running the Heun EDM sampler does not produce visually pleasing samples when using the true conditional scores directly. We hypothesize that this is due to fact that during training, the model has never seen images where some pixels are clean and others are noisy and thus, it fails to generalize to these cases. We leave investigation of this to future work but note that training directly on such augmented clean/noisy images may alleviate this issue.

# E   PROOFS

## E.1   DISCO OBJECTIVE

Let $p(t)$ be a prior distribution over a "time" parameter[6] $t \in [0, T]$, let $p_d$ denote the data distribution, and let $\alpha : [0, T] \to \mathbb{R}_{>0}$ and $\sigma : [0, T] \to \mathbb{R}_{>0}$ be positive functions of time. Given the distributions $p(t, \mathbf{x}, \mathbf{x}_t) := p(t)p_d(\mathbf{x})p_t(\mathbf{x}_t \,|\, \mathbf{x})$ with $p_t(\mathbf{x}_t \,|\, \mathbf{x}) := \mathcal{N}(\mathbf{x}_t; \alpha(t)\mathbf{x}, \sigma(t)^2 I)$ and $q(t, \mathbf{x}, \mathbf{x}_t) := p(t)p_t(\mathbf{x}_t)p_0(\mathbf{x} \,|\, \mathbf{x}_t)$ with $p_t(\mathbf{x}_t) = \int p_t(\mathbf{x}_t \,|\, \mathbf{x})p_d(\mathbf{x}) \, d\mathbf{x}$ and

$$p_0(\mathbf{x} \,|\, \mathbf{x}_t) = \frac{p_0(\mathbf{x}_t \,|\, \mathbf{x})p_d(\mathbf{x})}{p_0(\mathbf{x}_t)},$$

we will show that the *DISCO Loss*

$$\mathcal{L}_{\text{DISCO}}(\boldsymbol{\theta}) := \mathbb{E}_{q(t, \mathbf{x}, \mathbf{x}_t)}\left[\lambda(t)\|\nabla_{\mathbf{x}_t} \log p_0(\mathbf{x}_t \,|\, \mathbf{x}) - s_{\boldsymbol{\theta}}(\mathbf{x}_t)\|_2^2\right] \tag{14}$$

---

[6]We want to stress that it has only the meaning of time in diffusion models, while in DISCO it indexes a family of successively noisier proposal distributions.

is equivalent to

$$\mathcal{F}_{\text{mix}}(p_0 \,\|\, s_{\boldsymbol{\theta}}) = \mathbb{E}_{p(t)}\left[\lambda(t)\mathcal{F}_{p_t}(p_0 \,\|\, s_{\boldsymbol{\theta}})\right] = \mathbb{E}_{p(t)}\mathbb{E}_{p_t(\mathbf{x}_t)}\left[\lambda(t)\|\nabla_{\mathbf{x}_t}\log p_0(\mathbf{x}_t) - s_{\boldsymbol{\theta}}(\mathbf{x}_t)\|_2^2\right] \tag{15}$$

up to an additive constant independent of $\boldsymbol{\theta}$. As defined above, $p_0(\mathbf{x})$ is the slightly Gaussian-perturbed version of $p_d$ and is also called $p_d'$ in the main text.

*Proof.* We see that

$$\mathcal{L}_{\text{DISCO}}(\boldsymbol{\theta}) = \mathbb{E}_{q(t,\mathbf{x},\mathbf{x}_t)}\left[\lambda(t)\|\nabla_{\mathbf{x}_t}\log p_0(\mathbf{x}_t \,|\, \mathbf{x}) - s_{\boldsymbol{\theta}}(\mathbf{x}_t)\|_2^2\right]$$
$$= \mathbb{E}_{p(t)p_t(\mathbf{x}_t)}\left[\lambda(t)\,\mathbb{E}_{p_0(\mathbf{x} \,|\, \mathbf{x}_t)}\left[\|\nabla_{\mathbf{x}_t}\log p_0(\mathbf{x}_t \,|\, \mathbf{x}) - s_{\boldsymbol{\theta}}(\mathbf{x}_t)\|_2^2\right]\right]$$

We have

$$\mathbb{E}_{p_0(\mathbf{x} \,|\, \mathbf{x}_t)}\left[\|\nabla_{\mathbf{x}_t}\log p_0(\mathbf{x}_t \,|\, \mathbf{x}) - s_{\boldsymbol{\theta}}(\mathbf{x}_t)\|_2^2\right]$$
$$= \mathbb{E}_{p_0(\mathbf{x} \,|\, \mathbf{x}_t)}\left[\|\nabla_{\mathbf{x}_t}\log p_0(\mathbf{x}_t \,|\, \mathbf{x})\|_2^2 - 2\nabla_{\mathbf{x}_t}\log p_0(\mathbf{x}_t \,|\, \mathbf{x})^\top s_{\boldsymbol{\theta}}(\mathbf{x}_t) + \|s_{\boldsymbol{\theta}}(\mathbf{x}_t)\|_2^2\right]$$
$$= c_1 - 2\mathbb{E}_{p_0(\mathbf{x} \,|\, \mathbf{x}_t)}\left[\nabla_{\mathbf{x}_t}\log p_0(\mathbf{x}_t \,|\, \mathbf{x})\right]^\top s_{\boldsymbol{\theta}}(\mathbf{x}_t) + \|s_{\boldsymbol{\theta}}(\mathbf{x}_t)\|_2^2$$
$$= c_2 + \|\mathbb{E}_{p_0(\mathbf{x} \,|\, \mathbf{x}_t)}\left[\nabla_{\mathbf{x}_t}\log p_0(\mathbf{x}_t \,|\, \mathbf{x})\right] - s_{\boldsymbol{\theta}}(\mathbf{x}_t)\|_2^2$$

where $c_1, c_2$ are constants w.r.t. $\boldsymbol{\theta}$. We notice that

$$\mathbb{E}_{p_0(\mathbf{x} \,|\, \mathbf{x}_t)}\left[\nabla_{\mathbf{x}_t}\log p_0(\mathbf{x}_t \,|\, \mathbf{x})\right] = \int p_0(\mathbf{x} \,|\, \mathbf{x}_t)\nabla_{\mathbf{x}_t}\log p_0(\mathbf{x}_t \,|\, \mathbf{x})\,d\mathbf{x}$$
$$= \int p_0(\mathbf{x} \,|\, \mathbf{x}_t)\frac{\nabla_{\mathbf{x}_t}p_0(\mathbf{x}_t \,|\, \mathbf{x})}{p_0(\mathbf{x}_t \,|\, \mathbf{x})}\,d\mathbf{x}$$
$$= \int \frac{p_0(\mathbf{x}_t \,|\, \mathbf{x})p_0(\mathbf{x})}{p_0(\mathbf{x}_t)}\frac{\nabla_{\mathbf{x}_t}p_0(\mathbf{x}_t \,|\, \mathbf{x})}{p_0(\mathbf{x}_t \,|\, \mathbf{x})}\,d\mathbf{x}$$
$$= \int \frac{p_0(\mathbf{x})\nabla_{\mathbf{x}_t}p_0(\mathbf{x}_t \,|\, \mathbf{x})}{p_0(\mathbf{x}_t)}\,d\mathbf{x}$$
$$= \frac{1}{p_0(\mathbf{x}_t)}\int p_0(\mathbf{x})\nabla_{\mathbf{x}_t}p_0(\mathbf{x}_t \,|\, \mathbf{x})\,d\mathbf{x}$$
$$= \frac{1}{p_0(\mathbf{x}_t)}\nabla_{\mathbf{x}_t}\int p_0(\mathbf{x})p_0(\mathbf{x}_t \,|\, \mathbf{x})\,d\mathbf{x}$$
$$= \frac{1}{p_0(\mathbf{x}_t)}\nabla_{\mathbf{x}_t}p_0(\mathbf{x}_t)$$
$$= \nabla_{\mathbf{x}_t}\log p_0(\mathbf{x}_t)$$

and hence,

$$\|\mathbb{E}_{p_0(\mathbf{x} \,|\, \mathbf{x}_t)}\left[\nabla_{\mathbf{x}_t}\log p_0(\mathbf{x}_t \,|\, \mathbf{x})\right] - s_{\boldsymbol{\theta}}(\mathbf{x}_t)\|_2^2 = \|\nabla_{\mathbf{x}_t}\log p_0(\mathbf{x}_t) - s_{\boldsymbol{\theta}}(\mathbf{x}_t)\|_2^2$$

which implies that

$$\mathcal{L}_{\text{DISCO}}(\boldsymbol{\theta}) = \mathbb{E}_{p(t)p_t(\mathbf{x}_t)}\left[\lambda(t)\,\mathbb{E}_{p_0(\mathbf{x} \,|\, \mathbf{x}_t)}\left[\|\nabla_{\mathbf{x}_t}\log p_0(\mathbf{x}_t \,|\, \mathbf{x}) - s_{\boldsymbol{\theta}}(\mathbf{x}_t)\|_2^2\right]\right] + \text{const.}$$
$$= \mathbb{E}_{p(t)p_t(\mathbf{x}_t)}\left[\lambda(t)\,\|\nabla_{\mathbf{x}_t}\log p_0(\mathbf{x}_t) - s_{\boldsymbol{\theta}}(\mathbf{x}_t)\|_2^2\right] + \text{const.}$$
$$= \mathbb{E}_{p(t)}\left[\lambda(t)\mathcal{F}_{p_t}(p_0 \,\|\, s_{\boldsymbol{\theta}})\right] + \text{const.}$$

which concludes the proof. $\square$

## E.2 MULTISCALE DENOISING SCORE MATCHING

We show that the *multiscale denoising score matching (MDSM)* [Li et al., 2023] objective

$$\mathcal{L}_{\text{MDSM}}(\boldsymbol{\theta}) = \mathbb{E}_{p(t)p_d(\mathbf{x})p_t(\mathbf{x}_t \,|\, \mathbf{x})}\left[\lambda(t)\,\|\nabla_{\mathbf{x}_t}\log p_0(\mathbf{x}_t \,|\, \mathbf{x}) - s_{\boldsymbol{\theta}}(\mathbf{x}_t)\|_2^2\right] \tag{16}$$

has the minimizer $s_{\boldsymbol{\theta}}^*(\mathbf{x}_t) = \mathbb{E}_{p(t \,|\, \mathbf{x}_t)}\left[\frac{\sigma(t)^2}{\sigma(0)^2}\nabla_{\mathbf{x}_t}\log p_t(\mathbf{x}_t)\right]$ when $\lambda(t) = 1$ and $\alpha(t) = 1$ for all $t$ (variance exploding).

*Proof.* For convenience, we assume $\lambda(t) = 1$, as this can always be subsumed into the prior $p(t)$ without affecting the minimizer. Moreover, we assume $\alpha(t) = 1$. With $p(t, \mathbf{x}, \mathbf{x}_t) = p(t)p_d(\mathbf{x})p_t(\mathbf{x}_t \mid \mathbf{x})$, we denote with $p(\mathbf{x}_t)$ the marginal over $\mathbf{x}_t$ (not to be confused with $p_t(\mathbf{x}_t)$, which conditions on $t$). We have

$$\mathcal{L}_{\mathrm{MDSM}}(\boldsymbol{\theta}) = \mathbb{E}_{p(t)p_d(\mathbf{x})p_t(\mathbf{x}_t \mid \mathbf{x})} \left[ \|\nabla_{\mathbf{x}_t} \log p_0(\mathbf{x}_t \mid \mathbf{x}) - s_{\boldsymbol{\theta}}(\mathbf{x}_t)\|_2^2 \right] \tag{17}$$

$$= \mathbb{E}_{p(\mathbf{x}_t)p(t \mid \mathbf{x}_t)p_t(\mathbf{x} \mid \mathbf{x}_t)} \left[ \|\nabla_{\mathbf{x}_t} \log p_0(\mathbf{x}_t \mid \mathbf{x}) - s_{\boldsymbol{\theta}}(\mathbf{x}_t)\|_2^2 \right] \tag{18}$$

$$= \mathbb{E}_{p(\mathbf{x}_t)p(t \mid \mathbf{x}_t)} \left[ \|\mathbb{E}_{p_t(\mathbf{x} \mid \mathbf{x}_t)} [\nabla_{\mathbf{x}_t} \log p_0(\mathbf{x}_t \mid \mathbf{x})] - s_{\boldsymbol{\theta}}(\mathbf{x}_t)\|_2^2 \right] + \mathrm{const.} \tag{19}$$

where the last step follows the same argument as in Section E.1. With $R(\mathbf{x}_t, t) := \mathbb{E}_{p_t(\mathbf{x} \mid \mathbf{x}_t)} [\nabla_{\mathbf{x}_t} \log p_0(\mathbf{x}_t \mid \mathbf{x})]$ and repeating this argument, we see that

$$\mathbb{E}_{p(\mathbf{x}_t)p(t \mid \mathbf{x}_t)} \left[ \|R(\mathbf{x}_t, t) - s_{\boldsymbol{\theta}}(\mathbf{x}_t)\|_2^2 \right] = \mathbb{E}_{p(\mathbf{x}_t)} \left[ \|\mathbb{E}_{p(t \mid \mathbf{x}_t)} [R(\mathbf{x}_t, t)] - s_{\boldsymbol{\theta}}(\mathbf{x}_t)\|_2^2 \right] + \mathrm{const.} \tag{20}$$

where clearly, the minimizer is

$$s_{\boldsymbol{\theta}}^*(\mathbf{x}_t) = \mathbb{E}_{p(t \mid \mathbf{x}_t)} [R(\mathbf{x}_t, t)] \tag{21}$$

$$= \mathbb{E}_{p(t \mid \mathbf{x}_t)p_t(\mathbf{x} \mid \mathbf{x}_t)} [\nabla_{\mathbf{x}_t} \log p_0(\mathbf{x}_t \mid \mathbf{x})]. \tag{22}$$

Expanding $\nabla_{\mathbf{x}_t} \log p_0(\mathbf{x}_t \mid \mathbf{x}) = (\mathbf{x} - \mathbf{x}_t)/\sigma(0)^2$, we get

$$s_{\boldsymbol{\theta}}^*(\mathbf{x}_t) = \mathbb{E}_{p(t \mid \mathbf{x}_t)p_t(\mathbf{x} \mid \mathbf{x}_t)} \left[ \frac{\mathbf{x} - \mathbf{x}_t}{\sigma(0)^2} \right] = \mathbb{E}_{p(t \mid \mathbf{x}_t)} \left[ \frac{\mathbb{E}_{p_t(\mathbf{x} \mid \mathbf{x}_t)} [\mathbf{x}] - \mathbf{x}_t}{\sigma(0)^2} \right] \tag{23}$$

Via Tweedie's formula, we can express the posterior mean as $\mathbb{E}_{p_t(\mathbf{x} \mid \mathbf{x}_t)} [\mathbf{x}] = \mathbf{x}_t + \sigma(t)^2 \nabla_{\mathbf{x}_t} \log p_t(\mathbf{x}_t)$, and thus,

$$s_{\boldsymbol{\theta}}^*(\mathbf{x}_t) = \mathbb{E}_{p(t \mid \mathbf{x}_t)} \left[ \frac{\mathbf{x}_t + \sigma(t)^2 \nabla_{\mathbf{x}_t} \log p_t(\mathbf{x}_t) - \mathbf{x}_t}{\sigma(0)^2} \right] = \mathbb{E}_{p(t \mid \mathbf{x}_t)} \left[ \frac{\sigma(t)^2}{\sigma(0)^2} \nabla_{\mathbf{x}_t} \log p_t(\mathbf{x}_t) \right] \tag{24}$$

which concludes the proof. $\square$

This shows that the claim made in Li et al. [2023] that $s_{\boldsymbol{\theta}}^*(\mathbf{x})$ only learns $\nabla_{\mathbf{x}} \log p_0(\mathbf{x})$ is incorrect.

### E.3 TIME-INDEPENDENT DIFFUSION MODELS

We show that minimizing $\mathcal{L}_{\mathrm{DM}}$ with a *time-independent* score model $s_{\boldsymbol{\theta}}(\mathbf{x}_t)$, i.e.,

$$\mathcal{L}_{\mathrm{DM}}(\boldsymbol{\theta}) = \mathbb{E}_{t, \mathbf{x}_0, \mathbf{x}_t} \left[ \lambda(t) \|\nabla_{\mathbf{x}_t} \log p_t(\mathbf{x}_t \mid \mathbf{x}_0) - s_{\boldsymbol{\theta}}(\mathbf{x}_t)\|_2^2 \right], \tag{25}$$

leads to a minimizer $s_{\boldsymbol{\theta}}^*(\mathbf{x}_t) = \mathbb{E}_{p(t \mid \mathbf{x}_t)} [\nabla_{\mathbf{x}_t} \log p_t(\mathbf{x}_t)]$ when $\lambda(t) = 1$ and $\alpha(t) = 1$ for all $t$.

*Proof.* As the proof looks almost identical to Proof E.2, we will only briefly sketch it and refer the reader to Sun et al. [2025] for more details. With $R(\mathbf{x}_t, t) := \mathbb{E}_{p_t(\mathbf{x} \mid \mathbf{x}_t)} [\nabla_{\mathbf{x}_t} \log p_t(\mathbf{x}_t \mid \mathbf{x})]$, we again have that

$$s_{\boldsymbol{\theta}}^*(\mathbf{x}_t) = \mathbb{E}_{p(t \mid \mathbf{x}_t)} [R(\mathbf{x}_t, t)] = \mathbb{E}_{p(t \mid \mathbf{x}_t)p_t(\mathbf{x} \mid \mathbf{x}_t)} \left[ \frac{\mathbf{x} - \mathbf{x}_t}{\sigma(t)^2} \right] \tag{26}$$

Again via Tweedie's formula, we obtain

$$s_{\boldsymbol{\theta}}^*(\mathbf{x}_t) = \mathbb{E}_{p(t \mid \mathbf{x}_t)} \left[ \frac{\mathbf{x}_t + \sigma(t)^2 \nabla_{\mathbf{x}_t} \log p_t(\mathbf{x}_t) - \mathbf{x}_t}{\sigma(t)^2} \right] = \mathbb{E}_{p(t \mid \mathbf{x}_t)} [\nabla_{\mathbf{x}_t} \log p_t(\mathbf{x}_t)] \tag{27}$$

which concludes the proof. $\square$