# OpenReview forum: "Effective Diffusion-free Score Matching for Exact Conditional Sampling"
_auai.org/UAI/2025/Workshop/TPM — TPM 2025_

### Official Review · Reviewer_Wwot · 2025-06-16
**Accept**

**Rating:** 3

**Review:**

This paper proposes score matching models not based on diffusion enabling simple, principled conditional sampling. This paper is well written. The TPM community will find it relevant interesting.

One minor typo: “expect for one part.” on page 4

---

### Official Review · Reviewer_oixz · 2025-06-16
**interesting new technique with some overly strong claims.**

**Rating:** 3

**Review:**

The paper introduces a new loss to be optimized based on the q-weighted Fisher divergence instead of the Fisher divergence usually used in diffusion models. This then allows for tractably sampling asymptotically samples from the true distribution, which is not possible for diffusion models.
- The paper, however, has some issues that should be fixed. It is often mentioned (not always) that sampling is exact without mentioning that this only holds asymptotically. It would be good to be consistent and always mention the asymptotic behavior.
- Flow matching does also doe score matching without diffusion. While a recent paper by Tong et al. Is briefly mentioned in the appendix a more thorough comparison would be beneficial towards placing the paper in the literature, e.g. [1]

[1] Lipman, Yaron, et al. "Flow Matching for Generative Modeling." The Eleventh International Conference on Learning Representations.